# Immunopathogenesis of Atopic Dermatitis: Focus on Interleukins as Disease Drivers and Therapeutic Targets for Novel Treatments

**DOI:** 10.3390/ijms24010781

**Published:** 2023-01-02

**Authors:** Karolina Makowska, Joanna Nowaczyk, Leszek Blicharz, Anna Waśkiel-Burnat, Joanna Czuwara, Małgorzata Olszewska, Lidia Rudnicka

**Affiliations:** Department of Dermatology, Medical University of Warsaw, 02-008 Warsaw, Poland

**Keywords:** atopic dermatitis, biological treatment, biologics, cytokines, endotypes, interleukins, JAK inhibitors, pathogenesis, phenotypes, small molecules

## Abstract

Atopic dermatitis is a chronic, recurrent inflammatory skin disorder manifesting by eczematous lesions and intense pruritus. Atopic dermatitis develops primarily as a result of an epidermal barrier defect and immunological imbalance. Advances in understanding these pathogenetic hallmarks, and particularly the complex role of interleukins as atopic dermatitis drivers, resulted in achieving significant therapeutic breakthroughs. Novel medications involve monoclonal antibodies specifically blocking the function of selected interleukins and small molecules such as Janus kinase inhibitors limiting downstream signaling to reduce the expression of a wider array of proinflammatory factors. Nevertheless, a subset of patients remains refractory to those treatments, highlighting the complexity of atopic dermatitis immunopathogenesis in different populations. In this review, we address the immunological heterogeneity of atopic dermatitis endotypes and phenotypes and present novel interleukin-oriented therapies for this disease.

## 1. Introduction

Atopic dermatitis (AD) is a chronic, relapsing inflammatory skin disease with increasing worldwide prevalence and a significant impact on the patients’ quality of life [1]. AD is characterized by eczematous lesions showing typical, age-dependent distribution and intense pruritus [2]. The burden of AD is further associated with heterogeneous comorbidities (e.g., allergic respiratory diseases and autoimmune disorders), adverse effects of treatment (e.g., skin atrophy), and psychological distress resulting from social stigmatization [3,4].

The pathogenesis of AD involves genetic and environmental factors [2,5]. Impaired barrier function and immune dysregulation are two primary, interdependent phenomena responsible for the development of cutaneous inflammation.

Considering possible therapeutic benefits, special attention is given to explain the immunology of AD. The dominant role of the Th2 inflammatory axis is evident due to the high expression of IL-4 and IL-13 during the flares [6]. However, the heterogeneity of clinical pictures in different populations reflects the immunological complexity of AD and substantiates the view that it cannot be regarded as a uniform disease [7,8]. The molecular basis of these observations is subject to ongoing investigation to find new therapeutic targets and optimize current treatment strategies.

Despite certain gaps, the increasing knowledge of AD immunopathogenesis has already permitted elaboration of novel, highly efficient, and well-tolerated pharmaceuticals. These can be divided into biologics, including monoclonal antibodies selectively targeting proinflammatory cytokines, and small molecules inhibiting cellular downstream signaling to reduce the expression of a wider array of proinflammatory factors. Aside from the already registered drugs, new substances belonging to both of these groups are being studied.

The aim of this review is to discuss the role of interleukins in AD, evaluate population-dependent differences in AD immunopathogenesis associated with resulting therapeutic difficulties, and to outline new possibilities of systemic and topical treatment.

## 2. Materials and Methods

A literature search was performed using the PubMed database. The keywords used to perform the search were “atopic dermatitis” in different combinations with “interleukin *”, “inflame *”, “ethni *”, “biological treatment”, “JAK”, and “treatment”. Only articles in English were screened. Eligible articles retrieved from reference screening were included.

## 3. Interleukins in AD

Abnormal expression of interleukins plays a crucial role in the pathogenesis of AD [5]. Imbalance in the pro- and anti-inflammatory signals stimulates the vicious cycle of AD by triggering cutaneous inflammation, itch, and secondary impairment of the epidermal barrier [2,9]. Interleukins are produced by immunocompetent cells, such as T helper (Th) cells, Langerhans cells, and keratinocytes.

### 3.1. T Helper Cells

T helper cells (CD4+ T cells) are the primary source of interleukins and regulators of the immune response [10]. The major subtypes of CD4+ cells include Th1 cells, Th2 cells, Th17 cells, and Th22 cells.

Th1 cells are induced by IL-12 and IFN-γ secreted by dendritic cells and natural killer (NK) cells, most frequently upon recognition of pathogens by pattern recognition receptors [11]. Physiologically, Th1 cells are known to orchestrate cell-mediated responses by means of producing IL-2, IFN-γ, and TNF-α, which stimulates the function of cytotoxic T cells, natural killer cells, and macrophages. Upregulation of Th1-dependent cytokines is also seen in selected dermatological disorders such as psoriasis [12,13]. Despite the general domination of other immune pathways in AD, Th1 cells seem to play a considerable role in certain subgroups of AD patients, e.g., in those with intrinsic AD [6].

Th2 cells are the primary cytokines driving humoral responses [14]. Th2 cell function is potentiated by keratinocyte-derived molecules (IL-25, IL-33, and thymic stromal lymphopoietin, [TSLP]) and IL-4-producing cells such as basophils and innate lymphoid cells [15]. The main cytokines secreted by this subpopulation involve IL-4, IL-5, IL-13, and IL-31. Th2 cytokines drive AD severity by promoting cutaneous inflammation, inducing downregulation of skin barrier molecules (filaggrin, loricrin, and involucrin [16,17,18]) and IgE class switching. Recent discoveries showed that Th2 cytokines sensitize sensory neurons to pruritogens and, therefore, contribute to the development of chronic itch [19]. This process is mediated particularly by IL-31, a novel molecular target for AD treatment.

Th17 cells produce IL-17 and IL-22, while Th22 cells are only capable of secreting the latter. Physiologically, both IL-17 and IL-22 cells promote responses against bacteria, yeasts, and viruses [20]. In comparison to healthy controls, IL-17 expression is increased in the skin of some AD subgroups (e.g., in Asians), albeit not as prominently as in other inflammatory dermatoses such as psoriasis [13,21]. Expression of IL-22 is particularly marked in AD skin and correlates with disease severity. Th22 cells show limited expression in early childhood AD, but their progressive activation correlates with the changes in the morphology of skin lesions, which become more lichenified in older age groups. Significant upregulation of IL-22, but not IL-17 in the serum of AD patients, was reported to correlate with AD severity [22]. Increased production of IL-22 was also demonstrated in other dermatological conditions, including psoriasis [13,23], systemic sclerosis [24], and squamous cell carcinoma [25].

Furthermore, Toll-like receptors (TLR) signaling is an important response of the innate immunity, regulating Th1/Th17 and Th2 function in AD [26]. TLRs are activated by danger-associated molecular patterns (DAMPs), microbe-associated molecular patterns (MAMPs), pathogen-associated molecular patterns (PAMPs), and xenobiotic-associated molecular patterns (XAMPs) [27,28]. TLRs together with ILs are participating in the homeostasis of infections, autoimmune disorders, and cancers [28]. Moreover, TLRs and IL-1 receptors share the TIR domains and build a superfamily of versatile alarm mediators [27,28]. Th1/Th17 cells are mediated by activation of TLR2/TLR3/MAV in keratinocytes, TLR7/TLR8/TLR9 in dendritic cells, and TLR2/TLR4 in monocytes, which trigger pro-inflammatory cytokine production and T cell differentiation [26]. In contrast, Th2 response is initiated by impaired TLR2 function and leads to loss of skin barrier integrity [26], The disrupted innate immunity with Th2 dominance is important in the acute phase of AD.

### 3.2. Other Immunocompetent Cells Secreting Interleukins

Apart from lymphocytes, interleukins are also produced by keratinocytes, dendritic cells, and mast cells, among others [29]. Although these cellular responses are considered self-limiting and local, non-lymphocytic interleukin production is still regarded to significantly promote cutaneous inflammation.

As discussed above, keratinocytes form a part of the innate immune system [30], orchestrating ensuing patterns of cutaneous antimicrobial responses [30]. These cells play a pivotal role in the pathogenesis of AD [30]. Due to chemokine production, keratinocytes recruit other immune cells in the lesional sites, including dendritic cells, mast cells, eosinophils, and T cells [30,31]. Keratinocytes produce a cascade of pro-inflammatory factors, including thymic stromal lymphopoietin (TSLP), TNF-α, IL-1, IL-6, IL-8, IL-18, IL-23, IL-33, IL-36, as well as anti-inflammatory IL-38, which takes part in keratinocyte differentiation and counteracts the pro-inflammatory effect of IL-36 [30,32,33,34,35]. Exposure to *Staphylococcus aureus*, which typically dominates the microbiota of AD lesions, was demonstrated as one of the factors upregulating IL-36 with subsequent allergic reaction and increased synthesis of IL-4, IL-13, and IgE [36]. Furthermore, TSLP release from keratinocytes was associated with propensity for allergen sensitization, and IL-33 with IL-31 induction, itch, and downregulation of filaggrin and claudin-1 with subsequent disruption of the barrier function [37,38]. The combined presence of TNF-α and Th2 cytokines was shown to induce downregulation of epidermal differentiation complex proteins and stratum corneum lipids in an experimental model [39]. Upregulation of IL-8 both in the serum and AD lesions was shown to correlate with disease severity, probably by means of increased chemoattraction of immunocompetent antigen-presenting cells in the Th2 cytokine milieu [40].

Dendritic cells are capable of secreting IL-23, IL-25, IL-29, and IL-31 [29,41,42,43,44]. IL-23 helps to preserve IL-17 production by Th cells, while IL-25 stimulates IL-4 and IL-13 production by Th2 cells [41,43]. IL-29 is considered a type three interferon upregulating protective anti-viral responses [44]. Aside from the abovementioned role in chronic pruritus, IL-31 also activates IL-6, IL-16 and IL-32 production, as well as acts on chemotaxis of monocytes, T cells, and polymorphonuclear cells [42].

Mast cells produce a wide range of interleukins, including IL-1, 2, 3, 5, 6, 7, 8, 9, 13, 16, 17, and IL-33 [32,45]. As discussed above, the IL-1 family (including IL-36) and IL-33 play a particular role in driving cutaneous inflammation in AD [32]. Importantly, secretion of IL-9 supports T-cell survival and cross-activation of other mast cells [29]. IL-13 induces pro-inflammatory cytokine production, activates fibroblasts to synthesize collagen fibers, and supports B cell differentiation and IgE switching [29].

NK cells take part in IL-9, IL-13, IL-21, IL-22, and IL-31 secretion [29,46,47,48]. The role of IL-9 and IL-13 is similar to the role shown by mast cells [29,45]. IL-21 promotes B and T cell activation and differentiation as well as enhances NK cell activity [29,47].

Fibroblasts may secrete IL-1, IL-8, IL-11, and IL-38 [29,32,34]. Secreted IL-1 is strongly pro-inflammatory and leads to lymphocyte activation and macrophage stimulation [32]. IL-8 targets neutrophils, basophils, macrophages, mast cells, and keratinocytes, causing superoxide and granule release, neutrophil chemotaxis, and angiogenesis [29]. On the other hand, IL-11 was shown to inhibit pro-inflammatory cytokine secretion, while the anti-inflammatory IL-38 enhances keratinocyte differentiation [29,34].

## 4. Phase-Dependent Differences in Cytokine Expression in AD

The morphology of acute AD lesions involves prominent erythema, edema, and exudation. Chronic lesions are lichenified, dry, and hyperpigmented [2,49]. Molecular phenomena underlying acute and chronic phases of AD are constantly studied to elucidate the influence of the immune system on the clinical picture of this disease. Initially, the acute phase was solely regarded as Th2-driven, while the chronic phase was attributed to the domination of Th1 response [50]. Indeed, acute lesions show an abundant lymphocytic infiltrate in the skin as well as increased expression of IL-4, IL-5, IL-13, IL-31, and IL-33, which is a hallmark of Th2 response. Notwithstanding, later studies showed that Th2 response is accompanied by simultaneous Th22 activation, and a lesser induction of Th17 markers [7]. Chronic skin lesions are characterized by further upregulation of Th2 and Th22 cytokine axes, and additionally with increased expression of Th1, but not Th17 markers [7].

## 5. Endotype-Phenotype Correlation

Up-to-date classification of AD encompasses the distinction of patients’ subgroups based on the phenotype and/or the endotype [6]. The former is a classical approach, in which the course and prognosis can be categorized based on the clinical features sometimes referred to as stigmata. The most common examples include total IgE serum concentration, xerosis, white dermographism, palmar hyperlinearity, and Dennie-Morgan folds [6]. This classification may be additionally based on clusters of common serum biomarkers, allergy type (immediate or delayed hypersensitivity reactions [51]), and skin barrier status [6].

Endotype classification is based on the underlying molecular mechanisms. The distinction of endotypes is a more contemporary approach and is essential for personalizing the treatment [6,52]. Optimally, AD phenotypes should be substantiated by identifying the underlying molecular endotype. For example, ichthyosis constituting minor Hanifin-Rajka criteria [53] for AD is now known to result from filaggrin loss-of-function mutations [6]. Figure 1 summarizes the current paradigm on the activation of major Th subpopulations in different phases and endotypes of AD.

### 5.1. Intrinsic vs. Extrinsic Atopic Dermatitis

AD can be divided into intrinsic and extrinsic subtypes based primarily on the levels of total IgE [6]. The intrinsic subtype (10–40% of patients) is characterized by normal levels of total IgE, unaltered barrier function, female predominance, and generally lower disease severity compared to the extrinsic AD [6,54,55]. Common intrinsic AD stigmata include Dennie-Morgan folds and nasosinusal polyps [54]. The patients with intrinsic AD tend to present delayed rather than immediate hypersensitivity reactions [54]. Possibly, a deficiency of an epithelial peptide present in the skin and upper digestive tract, suprabasin (SBSN), may be responsible for increased nickel uptake and consequently an allergy to nickel in intrinsic patients [6,56,57]. On the molecular level, predominance of Th1, Th2, and Th17 responses results in high expression of IL-17A/IL-22, low expression of IL-4, IL-5, IL-13, and absence of specific IgE [6,54]. The extrinsic type (60–90%) is associated with eosinophilia, increased trans-epidermal water loss (TEWL), and filaggrin loss-of-function [6,54]. Patients with extrinsic AD often present with ichthyosis vulgaris and palmar hyperlinearity [54]. Th2 responses with high serum levels of specific IgE and elevated IL-4, IL-5, and IL-13 concentrations can be observed [6,54].

### 5.2. Ethnicity

Ethnicity and race are overlapping terms, with race based on inherited physical characteristics and ethnicity based on belonging to a group of ancestral origin [58]. The effect of ethnicity on the clinical picture of AD must be interpreted in the context of the quality and access to healthcare, socioeconomic status, and exposition to environmental factors (allergens, air pollution, chemical exposure) [59]. Notwithstanding, racial influence was found to be a strong factor determining the clinical picture of AD with significant differences among European American, Asian, and African American patients [7,59].

Compared to the European American population, Asian patients with AD tend to show increased Th17/Th22 responses, while the Th2 axis is similarly activated. This translates to high expression of IL-4, IL-5, IL-17, IL-19, and IL-22 [6,58,60]. Filaggrin mutations are less prevalent among Asian individuals in comparison to American Europeans [58]. Different cytokine profiles are thought to result in the distinct phenotype of Asian AD. This is reflected by the frequently observed psoriasiform reaction pattern involving epidermal hyperplasia and marked parakeratosis in histology. Clinically, adult patients of Japanese origin and those with a dark complexion more often develop prurigo-like lesions and follicular papules [61,62].

African American patients with AD present Th1/Th17 attenuation and Th2/Th22 skewing, which results in lower expression of IFN-γ and IL-17 than in patients of American European descent. This could possibly intensify the Th2-driven immunological imbalance resulting in a tendency for a more severe course of AD and IgE production [58,63]. Nevertheless, the prevalence of filaggrin loss-of-function seems to be lower in African Americans [6,58]. At the same time, *Staphylococcus aureus* colonization was found more frequently among African American children with AD [59]. The latter could be a risk factor for barrier dysfunction and allergen sensitization. Data in the literature further suggest that concomitant allergic contact dermatitis is less prevalent in dark skin phototypes, possibly due to less prominent Th1 reactions and lower cutaneous permeability [62,64].

Based on studies conducted in the United States [65] and the United Kingdom [66], the prevalence and severity of AD seem higher in African American children in comparison to European American children [67]. Masked erythema in patients with skin of color may contribute to a late diagnosis of AD [59,68]. This is also associated with the underestimation of AD severity in African American children when common scoring systems are used [59]. Finally, different evolution of skin lesions in dark phototypes should be considered, particularly the resolution of AD with post-inflammatory hypopigmentation [59,69].

### 5.3. Age

Children show age-dependent evolution of the underlying AD endotype. Initially, Th2 response predominates due to a lack of Th1 counterregulation, which translates to acute, exudative lesions [70]. Gradually, the Th22 axis becomes activated, reflecting a progressive tendency for lichenification [52]. Some studies of children with AD also identified a merged Th2/Th17-merged profile, associated with an increased IL-19 expression and possible psoriasiform inflammatory pattern [7]. Importantly, skewed immune responses with insufficient activation of Th1 axis in young children with AD make them particularly susceptible to infectious complications such as impetigo, eczema herpeticum, and molluscum contagiosum [61].

The evolving immunology of AD seems to underlie the changes in the morphology and distribution of skin lesions [2]. In infants, the lesions favor the face and extensor aspects of the extremities. Children over two years develop subacute lesions in the flexural folds. Finally, adolescents over 12 years and adults tend to present lichenified eczema of the flexures, face, hands, feet, and the back of the neck. Aside from the shift in immunological responses, this may also result from changes in the activity of sebaceous glands and the microbiome composition [52].

### 5.4. Gender

Sex hormones were shown to modulate immune responses. In general, male hormones such as testosterone tend to exert anti-inflammatory effects, whereas female hormones such as estrogen and progesterone are pro-inflammatory [71]. More specifically, testosterone seems to attenuate Th2 response, while estrogen and progesterone show propensity to downregulate Th1 response and exacerbate Th2-mediated inflammation. Additionally, estrogens probably affect the function of dendritic cells and type 2 innate lymphoid cells and enhance their function in allergic diseases [72,73]. This could potentially result in a higher prevalence and severity of AD in females, which is often transiently reflected during menstruation or pregnancy [74]. Nevertheless, epidemiological studies regarding the prevalence and severity of AD in men and women are conflicting. Data in the literature suggest that AD until the age of 65 years is more prevalent in females, whereas in the population over 65 years in males [71]. The immunological background of this observation is not fully elucidated. Importantly, overlap with other modulating factors and possible concomitant endotypes (e.g., ethnicity) should be considered.

### 5.5. Body-Mass Index

Obesity causes low-grade inflammation which contributes to the development of a wide range of comorbidities [75]. A recent systematic review of epidemiological data demonstrated that AD is associated with obesity, especially in infants [76,77,78]. Mechanistic data underlying these observations are scarce. One animal model study revealed that obesity causes significant upregulation of Th17 (IL-17A, IL-17F) and Th2 cytokines (IL-4 and IL-13), with the former axis activated more prominently [79]. These observations could partly underlie the possible cause of the increasing prevalence of AD in developed societies [80,81] where obesity has become a significant challenge for public health. Data in the literature suggest that weight reduction could positively affect treatment outcomes in AD [82]. Nevertheless, the attenuation of the Th2 axis with targeted treatments could potentiate the immune imbalance and result in a different Th17-mediated phenotype of AD as well as other negative sequelae (e.g., increased propensity to develop psoriasis) [83].

## 6. Genomics and Polymorphisms

The underlying genotype plays a pivotal role in the onset and clinical picture of AD. Single nucleotide polymorphisms (SNPs) of various genes were shown to be significantly associated with AD [84,85]. Several genome-wide association studies (GWAS) in European, Japanese, and Chinese populations helped to determine specific genetic susceptibility loci associated with AD development [85,86,87,88,89,90]. The loss-of-function mutations in the filaggrin (FLG) gene are considered the strongest genetic factor that increases the risk of AD [88,91,92,93]. The association between FLG null mutations and AD was first observed by Palmer et al. [91] FLG deficiency increases skin permeability, which facilitates the sensitization to environmental allergens and initiates the inflammatory cascade. This is reflected by the fact that patients with *FLG* loss-of-function mutation show higher levels of serum IgE [94,95]. As discussed above, FLG mutations are also a primary pathogenetic feature in extrinsic AD. Subsequent GWAS analyses revealed that apart from *FLG*, AD is also associated with other susceptibility loci, i.e., rs479844 located close to *OVOL1*, rs2897442 in *KIF3A* locus, and rs2164983 (at 19p13.2) located in an intergenic region between *ADAMTS10* and *ACTL9* [96].

*OVOL1* was implemented in the regulation of epidermal proliferation and differentiation. *KIF3A* encodes a subunit of kinesin-II complex, but the significance of polymorphism in this locus seems to result from its complex relationship with a cluster of cytokine and immune-related genes encoding IL-4 and IL-13. *ADAMTS10* is a gene encoding a member of ADAMTS zinc-dependent proteases which regulate extracellular matrix turnover and connective tissue remodeling. Lack of ADAMTS proteins was associated with spontaneous dermatitis presenting with epidermal thickening, dermal hypercellularity and extensive infiltration by immune cells in histopathology [97].

Different AD endotypes were also associated with SNPs in genes encoding cytokines. For example, the frequency of the IL-4Rα polymorphism C3223T and the IL-4 polymorphism C590T was shown to be higher in extrinsic AD than in intrinsic AD [98]. On the contrary, particular polymorphisms in IL-31 were typical for intrinsic, but not extrinsic AD [99]. Furthermore, higher risk of occurrence and increased persistence of AD was associated with amino acid change in the IL-6 receptor (IL-6R Asp358Ala; rs2228145), while polymorphisms in *IL5RA* were associated with a higher AD severity and eosinophil counts [100,101,102].

## 7. Diagnostics

The heterogeneity of AD endotypes and phenotypes translates to challenges in establishing distinct disease biomarkers. Furthermore, AD shares histological features with other eczematous disorders such as contact dermatitis. Therefore, the diagnosis of AD is currently based on sets of clinical criteria such as the Hanifin and Rajka criteria in adults or UK Working Party criteria in children [53,103]. Disease severity is assessed using measurement tools, e.g., the Eczema Area and Severity Index (EASI) and Scoring Atopic Dermatitis (SCORAD). However, the clinical applicability of the diagnostic criteria and measurement tools may be limited in certain subpopulations and be affected by inter-observer bias [104]. This reflects the need to identify reliable biomarkers facilitating the diagnosis and monitoring of patients with AD [104]. Among candidate diagnostic biomarkers are NOS2/iNOS, hBD-2, and MMP8/9 [92,105,106]. With respect to monitoring AD severity, one systematic review identified TARC/CCL17 to be reliable in both children and adults [107]. Other postulated candidates include SCCA2 [108,109], EDN [110], CTACK [92], MDC [111], LDH [112,113], and IL-18 [92,107]. For the assessment of treatment efficacy, biomarkers such as LDH [114], TARC, PARC, periostin, IL-22, eotaxin-1/3 [115], and IL-8 [116] could be used [92]. To date, however, none of the mentioned molecules has been implemented in clinical practice. Because of the mentioned heterogeneity of AD endotypes, future studies should be profiled to distinguish useful biomarkers in different patients’ subpopulations.

## 8. Therapeutic Challenges in Different AD Subtypes

The described heterogeneous expression of cytokines among certain populations is considered as a factor limiting classical treatment efficacy [52,61]. To date, the evidence regarding systemic treatments in different ethnical groups is scarce [62,117].

Traditionally, the treatment of refractory, moderate-to-severe AD relied on systemic anti-inflammatory agents. Recent advances in molecular sciences resulted in the introduction of new therapies targeting specific molecules involved in the pathogenesis of this disease (Figure 2).

### 8.1. Conventional Treatment of Atopic Dermatitis

Treatment of mild-to-moderate AD shares similar principles in different patients’ subpopulations. Combined with baseline emollient therapy, topical corticosteroids and calcineurin inhibitors are two primary groups of anti-inflammatory medications which enable sufficient control of AD in most individuals [118]. The mechanisms of action have been well investigated. Topical steroids are bound by their receptors and attenuate transcription of pro-inflammatory cytokines and simultaneously induce transcription of anti-inflammatory mediators. Topical calcineurin inhibitors reduce the activation of T cells by blocking the function of phosphorylase enzyme calcineurin. UV phototherapy (narrow-band UVB, UVA1) may be considered in patients with moderate-to-severe AD if the topical treatment is ineffective. Phototherapy is considered to alleviate cutaneous inflammation primarily by suppressing the function of antigen-presenting cells and T cells [119].

Immunosuppression can be used as an adjunct in patients with severe AD who are refractory to topical treatment. Several conventional immunosuppressants such as cyclosporine, methotrexate, azathioprine, and mycophenolate mofetil may be considered [120,121]. Immunosuppressants inhibit inflammatory response by affecting the interaction between antigen-presenting cells and T lymphocytes and reducing the populations of Th cells, synthesis of pro-inflammatory cytokines and histamine release from mast cells [122,123]. Active infections and malignancy should be ruled out before implementing any of the abovementioned medications [123]. Immunosuppressive therapy may be associated with serious adverse events (AEs), which vary based on the type of pharmaceutical, dose and the time of exposure. Symptoms such as hypertension, renal failure, hypertrichosis, and gingival hyperplasia (due to cyclosporine) or hepatotoxicity, bone marrow suppression, and pneumonitis (due to methotrexate) may lead to treatment discontinuation [123].

### 8.2. Novel Treatments of AD

#### 8.2.1. Biologics

Biologics are therapeutic agents obtained by means of biomedical engineering [124]. They are characterized by a relatively high molecular weight and complex structure. The use of biologics in dermatology is primarily constricted to monoclonal antibodies [125]. These molecules are characterized by high specificity towards a single molecular target, for example, interleukin or its receptor (Table 1). In this regard, they are well suited for personalized treatment adjusted to the immunological profile of AD endotypes. Nonetheless, the benefit-risk ratio should be always considered in the systemic treatment of AD, especially in pediatric and elderly patients. The burden of AD is significant, yet the disease is not immediately life-threatening. Therefore, safety measures should be taken to avoid threats of precision medicine such as the elevated risk of adverse events and drug interactions [4].

##### Dupilumab—An IL-4/IL-13 Inhibitor

Dupilumab is a recombinant monoclonal antibody against the IL-4 receptor, which is registered for use in children (over 6 months old in the United States and over 6 years old in Europe) and adults. The favorable safety profile and efficacy (reduction of body surface area [BSA], EASI, and IGA) were observed regardless of age, sex, race, and ethnicity [126]. AEs occurred in 13.5% of patients and involved mainly conjunctivitis and arthralgia. Of note, the introduction of dupilumab was found to aggravate lymphoma progression in most patients [127]. Therefore, a differential diagnosis should be performed, especially in middle-aged patients with a new onset of erythroderma [61,127].

Deng et al. [128] assessed the efficacy of dupilumab in patients treated for AD with concomitant palmo-plantar dermatitis. Reduced levels of eosinophils and IgE were found to be associated with a significant improvement in the severity of the disease. Furthermore, Shan et al. [129] highlighted the potential role of dupilumab in the treatment of AD with concomitant cheilitis.

High efficacy of dupilumab results from the primary role of Th2 cytokines in all AD endotypes. Nevertheless, recalcitrant cases are also seen, which suggests a higher complexity of particular AD subpopulations and simultaneous triggering of cutaneous inflammation by other cytokines. A recently published analysis of dupilumab nonresponders reinforces this hypothesis by suggesting that treatment efficacy is influenced by factors such as age, sex, race, ethnicity, and geographic areas [130].

##### Tralokinumab—An IL-13 Inhibitor

Tralokinumab is a human monoclonal antibody inhibiting IL-13 [131]. In a phase IIb study by Wollenberg et al. [131], adults with AD were randomized to receive 45, 150, and 300 mg of subcutaneous tralokinumab or placebo every 2 weeks for 12 weeks with concomitant TCs. At week 12, there was a significant decrease from baseline in the EASI score in tralokinumab groups compared with the placebo group (adjusted mean difference, −4.94; 95% CI, −8.76 to −1.13; *p* = 0.01). The most common treatment-associated AEs were headaches and infection of the upper respiratory tract.

Recently, Wollenberg et al. [132] published the results of two 52-week, randomized, double-blind, multicentre, placebo-controlled phase III trials (ECZTRA 1 and ECZTRA 2), in which adults with moderate to severe AD were randomly assigned to subcutaneous tralokinumab 300 mg every 2 weeks or placebo. At week 16, more patients from the tralokinumab group in comparison to the placebo group achieved an IGA score of 0 or 1: 15.8% vs. 7.1% in ECZTRA 1 (difference 8.6%, 95% CI 4.1–13.1; *p* = 0.002) and 22.2% vs. 10.9% in ECZTRA 2 (11.1%, 95% CI 5.8–16.4; *p* < 0.001). EASI-75 was achieved by: 25.0% vs. 12.7% (12.1%, 95% CI 6.5–17.7; *p* < 0.001) and 33.2% vs. 11.4% (21.6%, 95% CI 15.8–27.3; *p* < 0.001) of the patients, respectively. AEs were reported in 76.4% and 61.5% of patients in the tralokinumab group in ECZTRA 1 and ECZTRA 2, respectively. In the placebo group, AEs were experienced by 77.0% and 66.0% of patients in ECZTRA 1 and ECZTRA 2, respectively. The most frequent AEs of tralokinumab included upper respiratory tract infection and conjunctivitis.

A review of the therapeutic potential of tralokinumab in the treatment of AD published by Kelly et al. [133] revealed improvements in disease severity measures (including IGA scores and EASI-75 scores), and in quality of life (including pruritus scores; sleep interference scores; DLQI; SCORing Atopic Dermatitis, SCORAD; Patient Oriented Eczema Measure; and The Short Form 36 Health Survey).

##### Lebrikizumab—An IL-13 Inhibitor

Lebrikizumab, a humanized monoclonal antibody, is another IL-13 antagonist which has been evaluated in two phase II clinical trials in patients with moderate to severe AD. In 2018, Simpson et al. [134] conducted a randomized, placebo-controlled, double-blind, phase II study in which adult patients with moderate-to-severe AD were required to use TCs twice daily and then were randomized to lebrikizumab 125 mg single dose, 250 mg single dose, and 125 mg every 4 weeks for 12 weeks or placebo every 4 weeks for 12 weeks. At week 12, EASI-50 was achieved by a greater number of patients treated with lebrikizumab 125 mg every 4 weeks (82.4%; *p* = 0.026) in contrast to placebo every 4 weeks (62.3%). No statistically significant improvement in EASI-50 was shown in both single-dose treatment groups compared to placebo. Adverse events occurred with similar frequency in all study groups (66.7% all lebrikizumab groups vs. 66.0% placebo). Lebrikizumab was well tolerated with no serious AEs.

Guttman-Yassky et al. [135] performed a double-blind, placebo-controlled, dose-ranging randomized phase II clinical trial on the use of lebrikizumab in moderate-to-severe AD. Adult patients were randomized to receive subcutaneous injections of lebrikizumab at the following doses: 125 mg every 4 weeks (250 mg loading dose [LD]), 250 mg every 4 weeks (500 mg LD), and 250 mg every 2 weeks (500 mg LD at baseline and week 2), or placebo every 2 weeks. Compared with placebo (EASI least squares mean percentage change, −41.1% ± 56.5), lebrikizumab groups showed dose-dependent, statistically significant improvement in the primary endpoint vs. placebo at week 16: 125 mg every 4 weeks (−62.3% ± 37.3, *p* = 0.02), 250 mg every 4 weeks (−69.2% ± 38.3, *p* = 0.002), and 250 mg every 2 weeks (−72.1% ± 37.2, *p* < 0.001). Lebrikizumab had a favorable safety profile, with the most common AEs including upper respiratory tract infections, nasopharyngitis, headache, injection site pain, and fatigue. Injection site reactions (affecting 1.9% in placebo group vs. 5.7% in all lebrikizumab groups), herpesvirus infections (3.8% vs. 3.5%), and conjunctivitis (0% vs. 2.6%) were not commonly reported.

In 2022, Zhang et al. [136] published a systematic review and meta-analysis of seven randomized controlled trials evaluating the use of two IL-13 inhibitors, tralokinumab and lebrikizumab, in adult patients with moderate-to-severe AD. Compared to the placebo, both lebrikizumab and tralokinumab had greater improvement in EASI score (mean difference −20.37, 95%CI −32.28, −8.47). Both inhibitors had acceptable safety profiles, but their use was associated with a higher risk of conjunctivitis than placebo.

Importantly, both tralokinumab and lebrikizumab target the Th2 axis. Therefore, as in the case of dupilumab, some patients might not adequately respond to therapy. Due to the limited time of observation, the phenomenon of insufficient response should be progressively evaluated across different populations.

##### Spesolimab—An IL-36 Inhibitor

Spesolimab is a monoclonal antibody targeting IL-36 receptor, developed for the treatment of generalized pustular psoriasis in adults. Bissonnette et al. [137] conducted the first multicenter, randomized, double-blind, placebo-controlled, phase IIa study to evaluate the use of spesolimab in adult patients with moderate-to-severe AD. Patients were randomly assigned to receive intravenous spesolimab 600 mg or placebo every 4 weeks. After 16 weeks, a decrease in EASI score was shown, 37.9% for spesolimab vs. 12.3% for placebo (adjusted mean difference −25.6%, *p =* 0.149). No safety concerns were raised regarding spesolimab therapy. As discussed above, IL-36 is a Th1-dependent molecule, whose upregulation was observed upon stimulation by IL-22 and IL-17 [138]. Therefore, patients with less prominent Th2 activation, i.e., with intrinsic AD or predominantly chronic, lichenified phenotype, might benefit the most from this treatment in the future.

##### Nemolizumab—An IL-31 Inhibitor

Nemolizumab is a humanized monoclonal antibody that inhibits IL-31 and significantly reduces pruritus [139]. It is approved in Japan for use in children (over the age of 13 years) and adults with insufficient control of itch associated with AD [140].

In a double-blind, phase III trial conducted by Kabashima et al. [141], patients with moderate-to-severe AD were randomized to receive subcutaneous nemolizumab 60 mg or placebo every 4 weeks up to week 16, with concomitant topical agents. At week 16, the Visual Assessment Scale (VAS) score changed by −42.8% in the nemolizumab group and −21.4% in the placebo group. The EASI score decreased by 45.9% with nemolizumab and 33.2% with placebo. The DLQI score of 4 or less was achieved by 40% of patients in the nemolizumab group and 22% in the placebo group. Reaction to injection (unspecified) occurred in 8% of patients treated with nemolizumab and in 3% placebo group. However, further efficacy and safety trials should be performed.

Considering the mechanism of action, nemolizumab seems preferable in patients in whom pruritus constitutes the primary symptom of AD [140]. As IL-31 is not a primary cytokine orchestrating the Th2 response, it is not likely to influence signs of acute inflammation to a similar extent as other biologics listed above.

##### Fezakinumab—An IL-22 Inhibitor

Fezakinumab is a monoclonal antibody against the IL-22 receptor. Guttman-Yassky et al. [142] performed a randomized, double-blind, placebo-controlled trial with intravenous fezakinumab in monotherapy administered every 2 weeks for 10 weeks. At week 12, the fezekinumab group showed a significantly higher decline in SCORAD score (21.6 [3.8] vs. 9.6; [4.2]; *p* = 0.029), which was also noticed at week 20 (27.4 [3.9] vs. 11.5 [5.1]; *p* = 0.010). The most frequently reported AEs were upper respiratory tract infections. No further clinical trials of fezakinumab are currently ongoing.

Of note, the efficacy of fezakinumab was higher in patients with higher AD severity. This could reflect that progressive activation of the Th22 response correlates with the course of the disease. Most described AD endotypes share the common pathway of Th22 activation. However, considering the particularly important role of Th22 in Asian and African American AD, fezakinumab could be particularly efficient in those endotypes.

##### Tezepelumab—A Thymic Stromal Lymphopoietin Inhibitor

Tezepelumab is an anti-TSLP monoclonal antibody. Simpson et al. [143] conducted a randomized phase IIa clinical trial, in which patients were randomly assigned to receive TCs together with subcutaneous tezepelumab 280 mg or placebo every 2 weeks. At week 12, a higher percentage of patients treated with tezepelumab achieved an EASI-50 score (64.7%) than placebo (48.2%; *p* = 0.091). After 12 weeks, the treatment efficacy was insubstantial until improvement at week 16. The authors highlight the need for long-term trials to determine the efficacy of treatment. The occurrence of AEs was similar in both groups. To date, there is insufficient data to compare the expression of TSLP across AD subpopulations. TSLP is a primarily keratinocyte-derived molecule. Therefore, it is likely that the improvement of AD symptoms following successful TSLP inhibition would be similar in all groups.

##### Etokimab—An IL-33 Inhibitor

Etokimab is a monoclonal antibody acting as an IL-33 inhibitor. Chen et al. [144] conducted a proof-of-concept phase IIa study on a small group of adult patients with moderate-to-severe AD, who received a single intravenous 300 mg dose of etokimab. The EASI-50 score was achieved by 83% of patients, while EASI-75 was reached by 33%. Nearly one month after administration, the reduction in peripheral eosinophils was also noted. Etokimab was generally well tolerated with mild and transient AEs (e.g., headache, upper respiratory tract infections, urinary tract infections, peripheral swelling) assessed as unrelated to etokimab administration. IL-33 is another example of cytokine produced by the innate immune system, highlighting another possible molecular target in different AD endotypes. Nonetheless, another IL-33 antagonist, astegolimab, was not superior to placebo in a 16-week phase II randomized, placebo-controlled trial by Maurer et al. [145].

#### 8.2.2. Janus Kinase Inhibitors 

Janus Kinase (JAK) inhibitors are an emerging group of pharmaceuticals showing high efficacy in AD. Instead of targeting a single cytokine, JAK inhibitors inhibit downstream signaling and the production of a wider array of proinflammatory factors [146,147]. This highlights the possible therapeutic benefit in most AD endotypes, which results from the inhibition of several immune pathways. However, JAK inhibitors cannot be recognized as a one-size-fits-all treatment in AD, and recognition of the good-responder population is essential to optimize the benefit-risk ratio [4]. These small molecules are currently available for clinical use as oral or topical agents. Abrocitinib, upadacitinib, and baricitinib are administered orally [148,149], while ruxolitinib and deglocitinib are used in topical formulations [149].

##### Abrocitinib—Oral JAK 1 Inhibitor

Abrocitinib is a selective JAK 1 inhibitor accepted for the treatment of moderate-to-severe AD in adults in Europe, the United States, and Japan. This therapy can be used when systemic therapy (including biologics) has failed or is contraindicated.

In 2020, Simpson et al. [150] conducted a multicentre, double-blind, randomized, placebo-controlled, phase III trial involving patients ≥ 12 years of age. The participants were randomized to oral abrocitinib 100 mg and 200 mg, or placebo once daily for 12 weeks. IGA response was achieved in a higher number of patients treated with abrocitinib in comparison to placebo (24%, 44% vs. 8%, respectively; *p* = 0.0037, *p* < 0.0001). An EASI-75 response was significantly higher in the abrocitinib 100 mg group (40% vs. 12%; *p* < 0.0001) and abrocitinib 200 mg group (63% vs. 12%; *p* < 0.0001) compared to placebo. Serious AEs were reported in 3% of patients in the abrocitinib 100 mg group, 3% of patients in the abrocitinib 200 mg group, and 4% of patients in the placebo group. In adolescents and adults with moderate-to-severe AD, monotherapy with oral abrocitinib once daily was effective and well tolerated.

In another phase III, double-blind trial by Bieber et al. [151], patients were randomly assigned to the following groups: abrocitinib 200 mg, abrocitinib 100 mg, dupilumab group, and placebo group. At week 12, an IGA response was observed in 48.4% of patients in the 200 mg abrocitinib group, 36.6% in the 100 mg abrocitinib group, 36.5% in the dupilumab group, and 14.0% in the placebo group (*p* < 0.001 for both abrocitinib groups vs. placebo). Moreover, EASI-75 response was observed in 70.3%, 58.7%, 58.1%, and 27.1%, respectively (*p* < 0.001 for both abrocitinib groups vs. placebo). Regarding pruritus, a 200 mg dose of abrocitinib was superior to dupilumab. The most common AEs included nausea and acne.

##### Upadacitinib—Oral JAK 1 Inhibitor

Upadacitinib is a selective JAK 1 inhibitor approved in the United States and Europe for adults and children (aged ≥12 years) with AD uncontrolled by systemic drugs (including biologics) or with contraindications for such therapy.

The treatment with oral upadacitinib as monotherapy or with topical corticosteroids in adolescents and adults with moderate to severe AD was found effective in several multicenter, randomized trials [152,153]. In a randomized, double-blind, placebo-controlled, phase III trial by Reich et al. [152], ≥12 years old patients were enrolled to receive TCs with upadacitinib 15 mg, upadacitinib 30 mg, or placebo, once daily for 16 weeks. Validated IGA-AD response was achieved in a higher number of patients treated with upadacitinib compared to placebo (40%, 59% vs. 11%, respectively). An EASI-75 score was achieved in a significantly higher number of patients in the upadacitinib 15 mg (65%) and the upadacitinib 30 mg (77%) groups than in the placebo group (26%). The most common AEs (≥5% in any treatment group) were acne, nasopharyngitis, upper respiratory tract infection, oral herpes, elevation of blood creatine phosphokinase levels, and headache.

Guttman-Yassky et al. [153] published the results from two replicate double-blind, randomized controlled phase III trials (MEASURE UP 1 and MEASURE UP 2) in which patients aged ≥12 years were randomized to receive upadacitinib 15 mg, upadacitinib 30 mg, or placebo for 16 weeks. Higher scores in validated IGA-AD response were achieved in both upadacitinib groups in contrast to the placebo group (48%, 62%, and 8% in the MEASURE UP 1, respectively; 39%, 52%, and 5% in the MEASURE UP 2, respectively). The EASI-75 score was higher in the upadacitinib 15 mg (70%, 60%) and upadacitinib 30 mg (80%, 73%) groups than the placebo group (16%, 13%) in MEASURE UP 1 and MEASURE UP 2, respectively. Both upadacitinib doses were well tolerated with most frequent AEs including acne, upper respiratory tract infection, nasopharyngitis, and headache.

Compared to dupilumab, upadacitinib demonstrated higher efficacy. An EASI-75 score was achieved by a higher number of patients treated with upadacitinib than dupilumab (71% vs. 61%). Furthermore, the reduction in pruritus was higher in the upadacitinib group (55% vs. 36%) [154].

##### Baricitinib—Oral JAK 1/2 Inhibitor

Baricitinib is the first-generation JAK 1/2 inhibitor blocking cytokine signaling, including IL-4, IL-5, and IL-13, registered for the treatment of moderate-to-severe AD in the United States and Europe.

Simpson et al. [155] compared results from two randomized, double-blind monotherapy, phase III trials (BREEZE-AD1 and BREEZE-AD2). Adults with moderate-to-severe AD were required to use a once-daily placebo or baricitinib 1 mg, 2 mg, and 4 mg for 16 weeks. At week 16, the end point of validated IGA-AD was achieved by a higher number of patients treated with 4 mg and 2 mg barticitinib than placebo in BREEZE-AD1 (baricitinib 4 mg, 16.8%, *p* < 0.001; 2 mg, 11.4%, *p* < 0.05; 1 mg, 11.8%, *p* < 0.05; placebo, 4.8%), and BREEZE-AD2 (baricitinib 4 mg, 13.8%, *p* = 0.001; 2 mg, 10.6%; *p* < 0.05; 1 mg, 8.8%, *p* = 0.085; placebo, 4.5%). The most frequently reported AEs were nasopharyngitis and headache.

Results from a randomized monotherapy phase III trial in the United States and Canada (BREEZE-AD5) conducted by Simpson et al. [156] confirmed that baricitinib (1 mg or 2 mg) is effective in the therapy of patients with moderate-to-severe AD. New findings regarding safety were not observed in this study.

Results from the BioDay Registry published by Boesjes et al. [157] showed that the probability of achieving EASI ≤ 7 and NRS pruritus ≤ 4 using baricitinib for 16 weeks is 29.4% (range, 13.1–53.5) and 20.5% (range, 8.8–40.9), respectively. AEs included nausea (11.8%), urinary tract infection (9.8%), and herpes simplex infection (7.8%).

The risk of serious infections was found to be similar in patients with AD treated with baricitinib compared with a placebo [158].

##### Tofacitinib—Topical JAK Inhibitor

Tofacitinib is a JAK 1 and JAK 3 inhibitor. In a phase IIa, randomized, double-blind, placebo-controlled trial by Bissonette et al. [159], adult patients were randomized to 2% tofacitinib or placebo ointment twice daily. EASI score change was significantly higher for tofacitinib (−81.7%) vs. placebo (−29.9%; *p* < 0.001). The occurrence of AEs was higher in the placebo group than in the tofacitinib group. Further investigations are necessary to assess the safety profile of tofacitinib in light of the U.S. Food and Drug Administration warning.

##### Ruxolitinib—Topical JAK Inhibitor

Topical ruxolitinib is a JAK 1 and JAK 2 inhibitor with rapid and sustained antipruritic and anti-inflammatory effects [160]. Efficacy of ruxolitinib is compared to triamcinolone, but sparing the AEs observed during long-term TCs use [160]. In patients treated with topical ruxolitinib with up to 20% BSA affected by AD, drug plasma concentrations were not reached to the extent that induced AEs commonly associated with oral JAK inhibitors [161,162]. Topical ruxolitinib was approved in 2021 by the U.S. FDA for the short-term treatment of mild-to-moderate AD in patients over 12 years old.

A phase II, randomized, dose-ranging, placebo- and active-controlled study performed by Kim et al. [163] evaluated the effects of ruxolitinib cream on itch and quality of life. Patients with AD were enrolled to receive ruxolitinib cream 1.5% twice daily, 1.5% once daily, 0.5% once daily, 0.15% once daily, placebo twice daily, or triamcinolone cream (0.1% twice daily for 4 weeks, then placebo for 4 weeks). Overall, pruritus was decreased in 42.5% of patients who applied 1.5% ruxolitinib cream twice daily, which was also associated with an improvement in the patients’ quality of life. No serious ruxolitinib-related AEs were reported.

Papp et al. [164] presented the results of two phase III, randomized, double-blind studies, in which patients with AD were randomized to twice-daily 0.75% ruxolitinib cream, 1.5% ruxolitinib cream, or placebo cream for 8 weeks. At the end of the study, a significantly higher number of patients treated with 0.75% ruxolitinib cream (50.0% and 39.0% in these studies, respectively) and 1.5% ruxolitinib cream (53.8% and 51.3%) achieved IGA treatment success in comparison to placebo (15.1% and 7.6%, respectively; *p* < 0.0001). Furthermore, a greater itch reduction was noted in the ruxolitinib cream group compared to the placebo group. No clinically significant AEs were reported.

##### Delgocitinib—Topical JAK Inhibitor

Delgocitinib inhibits JAK 1, JAK 2, and JAK 3 as well as tyrosine kinase 2. The topical formulation is approved in Japan for adult patients with AD [165]. Nakagawa et al. [165] conducted a phase III, randomized, double-blind, placebo-controlled study and a subsequent open-label, long-term study to evaluate the efficacy and safety of delgocitinib ointment in pediatric patients. Participants were randomized to delgocitinib 0.25% ointment or placebo ointment used for 4 weeks followed by a 52-week extension period. The delgocitinib ointment group had significantly greater results than the control group (EASI score −39.3% vs. +10.9%, *p* < 0.001). Delgocitinib was well tolerated with the occurrence of mild AEs only.

## 9. Conclusions

The pathogenesis of AD is strictly associated with the imbalance in the interleukin network. This entails other molecular processes resulting in the development of AD lesions. Recent advances highlighted the heterogeneity of AD immunopathogenesis in different populations, which correlates with the heterogeneous clinical features of this disease. Tailoring the treatment to the endotypes of AD is a promising strategy that could limit the rates of nonresponders and reduce the worldwide burden of this disease. This can be achieved by optimizing the treatment using novel pharmaceuticals such as biologics and small molecules.

## Figures and Tables

**Figure 1 ijms-24-00781-f001:**
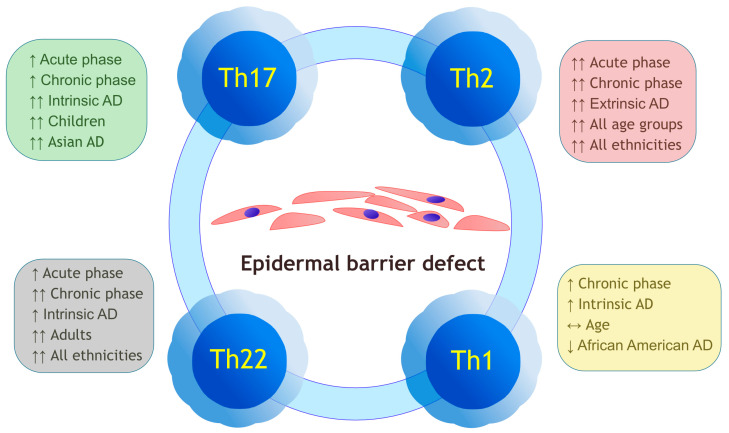
Current paradigm on the activation of major Th subpopulations in different phases and endotypes of AD. The immunological imbalance is interdependent with epidermal barrier defect, which produces the vicious cycle of AD. Legend: AD—atopic dermatitis, ↔—no effect, ↑—upregulation, ↑↑—significant upregulation, ↓—downregulation.

**Figure 2 ijms-24-00781-f002:**
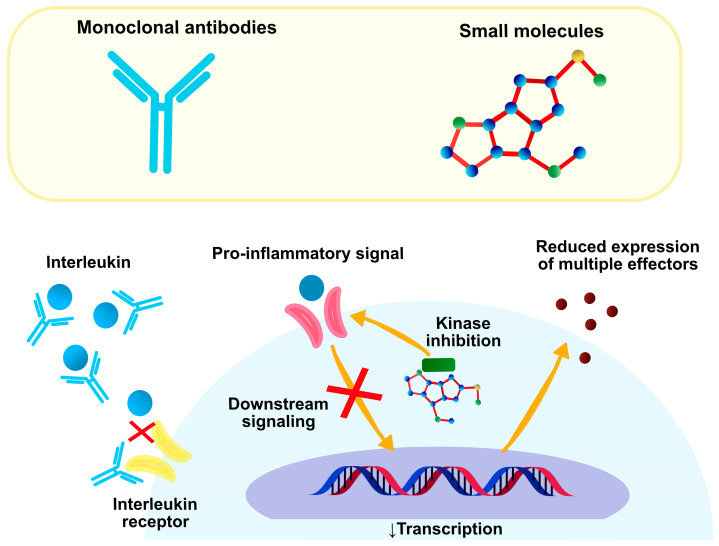
Mechanism of action of novel medications for AD. Monoclonal antibodies have large molecular weight and complex structure, which entails their extracellular function. They are characterized by highly specific inhibition of single interleukins or interleukin receptors. Small molecules act as intracellular kinase inhibitors, which limits downstream transduction of pro-inflammatory signals and subsequent transcription of a wide range of interleukins aggravating AD.

**Table 1 ijms-24-00781-t001:** Interleukin-targeting biological drugs for treatment of moderate-to-severe atopic dermatitis.

Interleukin (IL)	Cells Capable of IL Expression	Immune Function	Targeted Medications
IL-13	Th2 cells, T cells, NKT cells, mast cells, basophils, eosinophils	Promotion of B cell isotype switching; regulation of the antiparasitic response	Tralokinumab Lebrikizumab Dupilumab
IL-4	Normal T cells and B cells, cancerous B cells	Regulation of antibody production, inflammation, and effector T-cell response	Dupilumab
IL-22	Th17, Th22 and γδ T cells, activated NK cells	Prevention of tissue damage (activation of proliferative and anti-apoptotic pathways); regulation of the antimicrobial response	Fezakinumab
IL-31	Activated CD4+ Th2 cells, mast cells, monocytes, macrophages, dendritic cells	Induction of chemokine production by keratinocytes; modulation of eosinophil function; induction of itching sensation (by receptors on sensory neurons)	Nemolizumab
IL-33	Keratinocytes, macrophages, dendritic cells, fibroblasts, adipocytes, smooth muscle cells, endothelial cells, bronchial epithelium, osteoblasts, intestines	Activation of mast cells and basophils → overproduction of proinflammatory cytokines	Etokimab
IL-36	Keratinocytes, plasma cells, T-cells, macrophages and dendritic cells	Activation of pro-inflammatory pathways in response to tissue injury or infection; NF-κB activation; increasing Th-17 response	Spesolimab
TSLP	Fibroblasts, epithelial cells	Stimulation of Th2 response; promotion of antigen presenting cells maturation; promotion of eosynophil activity and chemotaxis; increasing the expression of IL-4, IL-5, and IL-13 in IL-33 stimulated human ILC2 cells	Tezepelumab

## Data Availability

No new data were created or analyzed in this study. Data sharing is not applicable to this article.

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
