# Peer review of "Immunopathogenesis of Atopic Dermatitis: Focus on Interleukins as Disease Drivers and Therapeutic Targets for Novel Treatments"

_ijms, 2023, doi:10.3390/ijms24010781_

Round 1
Reviewer 1 Report
The manuscript is a review of immunopathogenesis of atopic dermatitis with a focus on interleukins as disease drivers and therapeutic targets.
This is an interesting topic, the manuscript is clear and well organized. However, I would like to suggest some improvement before publication:
* Need to also consider IL-8 and TNF-alpha production by keratinocytes (section 3.2).
*Line 263 to 266: there is repletion of Figure 1 caption. Should be removed.
*Regarding AD treatment options, one need to also mention topical corticosteroids in the treatment of AD
*As an important part of the manuscript is focusing on the novel treatments of AD with a detailed inventory, I would recommend to reflect this in the manuscript title.
Author Response
Dear Reviewer,
Thank you for the time and effort dedicated to revise our manuscript. We updated the article according to your suggestions. Please find the point-by-point responses to the remarks below:
The manuscript is a review of immunopathogenesis of atopic dermatitis with a focus on interleukins as disease drivers and therapeutic targets.
This is an interesting topic, the manuscript is clear and well organized. However, I would like to suggest some improvement before publication:
We thank the Reviewer for the kind words and respond to the following remarks below.
* Need to also consider IL-8 and TNF-alpha production by keratinocytes (section 3.2).
We thank the Reviewer for pointing out the missing molecules produced by keratinocytes. We updated the section about keratinocytes to include information about the production of TNF-alpha and IL-8 and their possible role in AD immunopathogenesis (lines 121, 129-133, respectively). Following the suggestion of the other Reviewer we also elaborated on the role of keratiocytes as innate immune cells to further substantiate their role in atopic dermatitis and cited additional suggested literature (lines 117-120)
*Line 263 to 266: there is repletion of Figure 1 caption. Should be removed.
We thank the Reviewer for this valuable remark and apologize for the unnecessary repetition of Figure 1 caption. We removed this section of the manuscript.
*Regarding AD treatment options, one need to also mention topical corticosteroids in the treatment of AD
We thank the Reviewer for pointing out this important gap in the manuscript. We changed the section heading from 'Classic immunosuppressive treatment' to 'Conventional treatment of atopic dermatitis' and included the information on topical steroids, as well as topical calcineurin inhibitors and phototherapy (currently section no. 8.1., lines 362-376) to more fully present the therapeutic options in atopic dermatitis based on disease severity and other conditions.
*As an important part of the manuscript is focusing on the novel treatments of AD with a detailed inventory, I would recommend to reflect this in the manuscript title.
We thank the Reviewer for this valuable remark. We updated the title of the manuscript to be as follows:
Immunopathogenesis of Atopic Dermatitis: Focus on Interleukins as Disease Drivers and Therapeutic Targets for Novel Treatments
Reviewer 2 Report
Dear Authors
Thank you very much for your manuscript submission. The topic of the review is interesting; however, this manuscript needs a "Major Revision" as below:
1. It is recommended to read and add the following interesting papers to References section of the manuscript to have fruitful manuscript:
Overlapping Features of Psoriasis and Atopic Dermatitis: From Genetics to Immunopathogenesis to Phenotypes. Int J Mol Sci. 2022 May 15;23(10):5518. doi: 10.3390/ijms23105518. PMID: 35628327; PMCID: PMC9143118.
The JAK/STAT Pathway and Its Selective Inhibition in the Treatment of Atopic Dermatitis: A Systematic Review. J Clin Med. 2022 Jul 29;11(15):4431. doi: 10.3390/jcm11154431. PMID: 35956047; PMCID: PMC9369061.
Immunological Pathomechanisms of Spongiotic Dermatitis in Skin Lesions of Atopic Dermatitis. Int J Mol Sci. 2022 Jun 15;23(12):6682. doi: 10.3390/ijms23126682. PMID: 35743125; PMCID: PMC9223609.
Atopic Dermatitis Pathogenesis: Lessons From Immunology. Dermatol Pract Concept. 2022 Jan 1;12(1):e2022152. doi: 10.5826/dpc.1201a152. PMID: 35223190; PMCID: PMC8824231
Atopic dermatitis: an expanding therapeutic pipeline for a complex disease. Nat Rev Drug Discov. 2022 Jan;21(1):21-40. doi: 10.1038/s41573-021-00266-6. Epub 2021 Aug 20. PMID: 34417579; PMCID: PMC8377708.
Keratinocytes: innate immune cells in atopic dermatitis. Clin Exp Immunol. 2021 Jun;204(3):296-309. doi: 10.1111/cei.13575. Epub 2021 Feb 15. PMID: 33460469; PMCID: PMC8119845.
2. It is recommended to add a subtitle entitled "Diagnostics" to your manuscript. In this regard, please do read and add the following paper to References section of the manuscript to have fruitful review:
Atopic Dermatitis: Striving for Reliable Biomarkers. J Clin Med. 2022 Aug 9;11(16):4639. doi: 10.3390/jcm11164639. PMID: 36012878; PMCID: PMC9410433.
3. In section "5. Endotype-phenotype correlation"; it is recommended to add subtitles including "Age", "Gender" and "Body Mass Index".
4. It is recommended to add a subtitle entitled "Genomics and Polymorphisms". This section is very important for your manuscript.
5. As you know, interleukins and Toll-like receptors are sisters. They have pivotal roles in immune responses. In this regard, it is recommended to read and add the following papers to the References section of the manuscript:
The Role of Toll-Like Receptors in Skin Host Defense, Psoriasis, and Atopic Dermatitis. J Immunol Res. 2019 Nov 14;2019:1824624. doi: 10.1155/2019/1824624. PMID: 31815151; PMCID: PMC6877906.
The Interleukin-1 (IL-1) Superfamily Cytokines and Their Single Nucleotide Polymorphisms (SNPs). J Immunol Res. 2022 Mar 26;2022:2054431. doi: 10.1155/2022/2054431. PMID: 35378905; PMCID: PMC8976653.
Toll-Like Receptors: General Molecular and Structural Biology. J Immunol Res. 2021 May 29;2021:9914854. doi: 10.1155/2021/9914854. PMID: 34195298; PMCID: PMC8181103.
Author Response
Dear Reviewer,
Thank you for the time and effort dedicated to revise our manuscript. We updated the article according to your suggestions. Please find the point-by-point responses to the included remarks below:
Dear Authors
Thank you very much for your manuscript submission. The topic of the review is interesting; however, this manuscript needs a "Major Revision" as below:
We thank the Reviewer for the critical review of our work and reply to the suggestions below:
1. It is recommended to read and add the following interesting papers to References section of the manuscript to have fruitful manuscript:
Overlapping Features of Psoriasis and Atopic Dermatitis: From Genetics to Immunopathogenesis to Phenotypes. Int J Mol Sci. 2022 May 15;23(10):5518. doi: 10.3390/ijms23105518. PMID: 35628327; PMCID: PMC9143118.
The JAK/STAT Pathway and Its Selective Inhibition in the Treatment of Atopic Dermatitis: A Systematic Review. J Clin Med. 2022 Jul 29;11(15):4431. doi: 10.3390/jcm11154431. PMID: 35956047; PMCID: PMC9369061.
Immunological Pathomechanisms of Spongiotic Dermatitis in Skin Lesions of Atopic Dermatitis. Int J Mol Sci. 2022 Jun 15;23(12):6682. doi: 10.3390/ijms23126682. PMID: 35743125; PMCID: PMC9223609.
Atopic Dermatitis Pathogenesis: Lessons From Immunology. Dermatol Pract Concept. 2022 Jan 1;12(1):e2022152. doi: 10.5826/dpc.1201a152. PMID: 35223190; PMCID: PMC8824231
Atopic dermatitis: an expanding therapeutic pipeline for a complex disease. Nat Rev Drug Discov. 2022 Jan;21(1):21-40. doi: 10.1038/s41573-021-00266-6. Epub 2021 Aug 20. PMID: 34417579; PMCID: PMC8377708.
Keratinocytes: innate immune cells in atopic dermatitis. Clin Exp Immunol. 2021 Jun;204(3):296-309. doi: 10.1111/cei.13575. Epub 2021 Feb 15. PMID: 33460469; PMCID: PMC8119845.
We thank the Reviewer for providing us with interesting articles related to our manuscript. We included all the suggested articles in the bibliography. The new references (listed in the order provided by the Reviewer) are 13, 147, 51, 8, 4, and 30. Furthermore, we elaborated on certain parts of the manuscript basing on the information contained in the newly added references (lines 117-120, 455-460, 649-651).
2. It is recommended to add a subtitle entitled "Diagnostics" to your manuscript. In this regard, please do read and add the following paper to References section of the manuscript to have fruitful review:
Atopic Dermatitis: Striving for Reliable Biomarkers. J Clin Med. 2022 Aug 9;11(16):4639. doi: 10.3390/jcm11164639. PMID: 36012878; PMCID: PMC9410433.
We thank the Reviewer for pointing out this missing part of the manuscript. We added the whole section on Diagnostics (Section 7, lines 332-340) and cited the provided interesting article (reference 104).
3. In section "5. Endotype-phenotype correlation"; it is recommended to add subtitles including "Age", "Gender" and "Body Mass Index".
We thank the Reviewer for suggesting fields to elaborate on the section 'Endotype-phenotype correlation'.
Since data on age-dependent changes of AD immunology were discussed in the section 5.3. entitled 'Children', we decided to change its title to 'Age' to better reflect the content of this subheading.
Furthermore, we added sections 5.4. and 5.5., i.e. 'Gender' and 'Body-mass index' to provide more data relevant to the topic of our review (lines 267-282 and lines 284-296, respectively).
4. It is recommended to add a subtitle entitled "Genomics and Polymorphisms". This section is very important for your manuscript.
We thank the Reviewer for this excellent remark. We added a whole section entitled 'Genomics and polymorphisms' to the manuscript (section 6, lines 299-330).
5. As you know, interleukins and Toll-like receptors are sisters. They have pivotal roles in immune responses. In this regard, it is recommended to read and add the following papers to the References section of the manuscript:
The Role of Toll-Like Receptors in Skin Host Defense, Psoriasis, and Atopic Dermatitis. J Immunol Res. 2019 Nov 14;2019:1824624. doi: 10.1155/2019/1824624. PMID: 31815151; PMCID: PMC6877906.
The Interleukin-1 (IL-1) Superfamily Cytokines and Their Single Nucleotide Polymorphisms (SNPs). J Immunol Res. 2022 Mar 26;2022:2054431. doi: 10.1155/2022/2054431. PMID: 35378905; PMCID: PMC8976653.
Toll-Like Receptors: General Molecular and Structural Biology. J Immunol Res. 2021 May 29;2021:9914854. doi: 10.1155/2021/9914854. PMID: 34195298; PMCID: PMC8181103.
We thank the Reviewer for suggesting new, valuable references. We included all the articles in the bibliography. The new references (listed in the order provided by the Reviewer) are 26, 28, and 27. Furthermore, we elaborated on the role of TLRs in section 3.1. (lines 99-110).
Round 2
Reviewer 2 Report
No Revision is done. I think you have uploaded the previous version of your manuscript by the mistake. Please do highlight the revised sections by Yellow color within the revised manuscript.
Round 3
Reviewer 2 Report
Accept